# Evaluation of the SteamVR Motion Tracking System with a Custom-Made Tracker

**Marcin Maciejewski** 

Institute of Optoelectronics, Military University of Technology, gen. Sylwestra Kaliskiego 2, 00-908 Warsaw, Poland; marcin.maciejewski@wat.edu.pl; Tel.: +48-26-183-96-27

**Abstract:** The paper presents the research of the SteamVR tracker developed for a man-portable air-defence training system. The tests were carried out in laboratory conditions, with the tracker placed on the launcher model along with elements ensuring the faithful reproduction of operational conditions. During the measurements, the static tracker was moved and rotated in a working area. The range of translations and rotations corresponded to the typical requirements of a shooting simulator application. The results containing the registered position and orientation values were plotted on 3D charts which showed the tracker's operation. Further analyses determined the values of the systematic and random errors for measurements of the SteamVR system operating with a custom-made tracker. The obtained results with random errors of 0.15 mm and 0.008° for position and orientation, respectively, proved the high precision of the measurements.

**Keywords:** motion capture; SteamVR tracking; accuracy assessment

## 1. Introduction

Virtual reality (VR) systems have been gaining popularity in recent years due to the development of virtual technologies. They have become cheaper, the image visualisation possibilities have increased, and motion controllers more accurately transfer the user's actual actions to the virtual world. All this means that the use of virtual reality systems for applications much more advanced than those commonly associated with entertainment is being considered more often [1–3]. To name but a few industries, virtual reality is used in medicine [4,5], sport [6], and in military applications [7], where the typical use is for shooting simulators [8,9]. Some simulators display the image on a large screen but, in immersive solutions, the virtual world image is shown on a head-mounted display (HMD). Proper training in a virtual environment ensures cost reductions, increased training safety, as well as the possibility of creating many training scenarios.

The usefulness of a given training system, including a virtual one, is primarily determined by the transfer value, which means the possibility of applying the knowledge and skills acquired during training to a real environment and situations where they are used and needed [10]. The transfer value of virtual training simulators is influenced by several factors, such as immersion, i.e., the impression of being present in the virtual world, fidelity of the real-world representation, and the level of user acceptance of a given simulator [11–13]. For immersive shooting simulators, all these factors largely depend on the accuracy with which the application is able to reproduce the weapon's real movements. Therefore, it is extremely important to choose the right motion tracking system. Such a system provides real-time data on the position (coordinates x, y, z) and orientation (angles $\alpha$, $\beta$, $\gamma$) of the tracked object. It is essential that this data is continuously measured and delivered to the application without significant delays (below 20 ms [14]). It is equally important that the measurement is characterised by high precision (low random error value) and trueness (low systematic error value).

There are several types of motion tracking systems available on the market, the most popular of which are marker optoelectronic solutions, characterised by their high

accuracy [15–17]. In this respect, multi-camera systems are the best solution, [18] but their use entails the complicated installation and calibration of cameras and, above all, a high price. Single-camera systems are much simpler and cheaper [19], however they do not provide sufficient accuracy and, moreover, there may be problems with the camera's visibility of the tracked object, causing interruptions in tracking. An alternative approach to the systems considered above is the lighthouse solution [20,21], in which base stations located in the room are stationary, and the tracked object is equipped with a set of detectors. An example of this type of system is SteamVR (the full name is SteamVR Tracking), developed by Valve, which is used in commercial devices such as the HTC VIVE virtual reality set. SteamVR is an open system and can be used by other companies implementing it to track the devices they manufacture. Hardware development kits provided by Triad Semiconductors and Virtual Builds are helpful in implementing this technology [22].

This system was used in the developed PIORUN [23] shooting simulator (Polish MANPADS missile system). Initially, a commercial product from HTC VIVE was used as a tracked device (tracker). The SteamVR system operating with this tracker was tested [24], which proved the accuracy of the motion tracking was sufficient, but there were some problems with maintaining uninterrupted tracking of the launcher model. Accordingly, a proprietary design for the tracking device was developed, which was able to ensure continuous measurement of an object with dimensions as large as the PIORUN missile launcher.

The developed device was also initially tested [25], but in isolation from its intended application. In 95% of the tested orientations, the tracker was characterised by sufficient parameters. This means the measurement's precision, determined by a standard deviation of less than 1 mm for the position and $1°$ for the orientation, did not exceed the assumed limits. For other tracker orientations, however, worse values for the precision (up to 10 cm for position and $2°$ for orientation) and were recorded. These measurements were carried out under conditions intended to reproduce the operation of the HMD Designer simulation software, i.e., only one base station was used, and the measurements were carried out in an uncalibrated working area. Thus, the obtained results did not provide complete data on the parameters of the tracker when working in real conditions. Moreover, the previous measurement method does not verify the key feature of the tracker, which is the ability to continuously track launchers at a range of $0–360°$ for the azimuth and $0–60°$ for the elevation (based on the simulator's requirements). Earlier tests and simulations have confirmed that commercial devices, due to their shape and size, were obscured by the launcher and the user during operation. However, the developed tracker, according to the simulation results, should be free from this defect.

This paper describes the measurement setup and the analysis of the measurement results of the parameters for a custom-made tracker operating in a SteamVR system, optimised for tracking the PIORUN missile launcher model. The analysis provided the parameters of the tracker in real operating conditions for the shooting simulator. The tests also confirmed that the tracker was free from the limitations of commercial devices, i.e., the presence of orientations in which tracking was not possible.

A similar study to evaluate the performance of the SteamVR system and compatible devices was described in [21,26,27]. However, these studies concern commercial devices only. In turn, ref. [22] describes a custom-made device; however, the aspect of testing its parameters is only briefly presented. This article provides extensive research on a custom-made tracker compatible with the SteamVR tracking system.

## 2. Tracker and Measurement Setup

The subject of the research was the SteamVR system tracker equipped with a set of sensors (a photodetector with a signal conditioning system) receiving a predetermined sequence of signals from the base stations. By analysing these signals, the SteamVR system is able to determine the position and orientation of the tracker. The predicted random error of the measurement is inversely proportional to the distance between the sensors.

The tracker structure was designed using the CAD program; simulation software included in the SteamVR Hardware Development Kit (SteamVR HDK version 04558546), as well as analytical and simulation programs prepared by the author. The project was optimised so that the designed device was characterised by a high accuracy of the position and orientation measurements in the range of 0–360° azimuth and 0–60° elevation angles for the 2 × 2 m working area, which means:

1.  High precision, expressed by the standard deviation (*SD*) of the measurement results for each of the vector components:

    - position ($SD_X$, $SD_Y$, $SD_Z$) below 1 mm;
    - orientation ($SD_A$, $SD_B$, $SD_\Gamma$) below 0.1°.

2.  High trueness, which translates into the difference (Δ) between the mean measured value and the real value of the measured quantity:

    - each of the position components ($\Delta_X$, $\Delta_Y$, $\Delta_Z$) below 10 mm;
    - orientation ($\Delta_{orientation}$) below 1°.

The result of the design work was a model equipped with 24 sensors, which at the simulation stage had better parameters than commercial devices available on the market. Based on the model, a prototype of the device was made using 3D printing. Photos of the model and prototype are shown in Figure 1.

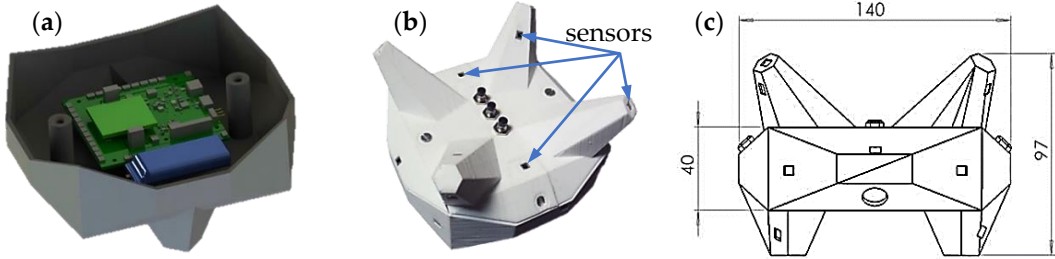

**Figure 1.** The lower part of the tracker model prepared for printing: with electronic components (**a**); a photo of the tracker prototype (**b**); and its dimensions (units—mm) (**c**).

The prototype was tested, which confirmed that the simulation program's predictions about the number of visible sensors were correct. Therefore, further tests were started, the aim of which were to determine the tracker's parameters in conditions typical for a shooting simulator.

A measuring setup was prepared in a laboratory room with dimensions of 3.6 by 4.5 m and a height of 1.98 to 2.15 m. Strong light sources and reflective elements that could distort the measurements were removed from the room. Two base stations (BS1 and BS2) of the SteamVR system were placed in opposite corners of the room at heights of 2.09 and 1.89 m. Their orientations were chosen so that they faced the centre of the room, and the inclination was set at 30° from the vertical. This is the minimum suggested value associated with the lower than recommended mounting height of the base stations. The distance between the base stations was over 5 m, which is more than the maximum distance for correct optical synchronisation. Therefore, wire synchronisation was used, which guaranteed uninterrupted work during the measurements. On the floor, in the centre of the room, nine measurement points were marked out, within a square with sides of 2 m. The distance between the points was 1 m. The distribution scheme for the measurement points is shown in Figure 2.

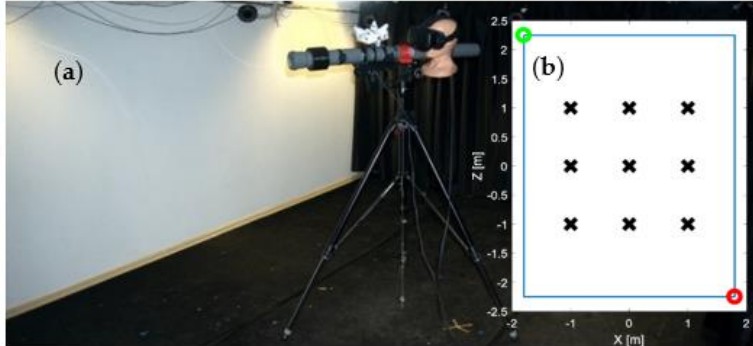

**Figure 2.** Photo of the laboratory room with a tracker and a model of the launcher on a tripod (**a**); and diagram of the distribution of the measuring points ("x" symbols) (**b**). The blue line marks the walls of the room, and the red and green circles symbolise the base stations.

At each of the nine points marked with the symbol "x", a tripod with a mounted launcher was placed sequentially. The tests were carried out for the launcher at a height of 1.6 m, which corresponds to the average height of a human arm. The geared tripod head Manfrotto 405 allowed the desired orientation of the launcher to be set. The tests were performed for the angles $\varphi$ = 0–360° (azimuth) in steps of 30° and $\theta$ = 0–80° (elevation) in steps of 20°. As a result, 65 different orientations were obtained, and tested at each of the nine points in the room, which give a total of 585 tested launcher positions. A greater concentration of measurements would be too time-consuming, due to the need to manually set the tripod for each measurement.

A model of the launcher was developed based on a PVC pipe with a diameter of 75 mm, on which the other elements, made by 3D printing, were mounted. To mount the tracker on the launcher model, a Picatinny rail (STANAG 2324) was used, which in the real launcher is used for mounting the scope. The model was equipped with a handle that allows it to be stably mounted on the tripod head (Figure 3). The model of the user's head, along with the HMD goggles from the HTC VIVE set, were mounted on the launcher. These are the goggles that will be used in the simulator application, so that real conditions can be reproduced more faithfully.

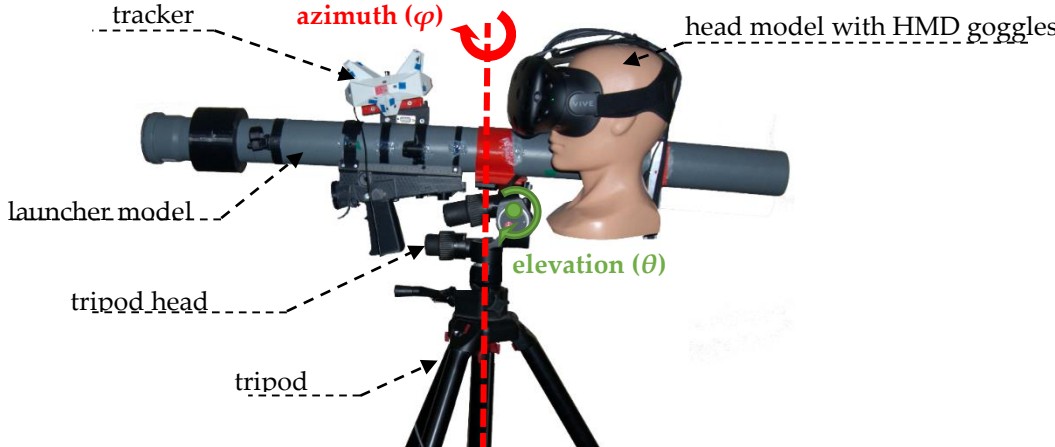

**Figure 3.** Photo of the launcher model with the mounted tracker and the user's head model.

Before the measurements were taken, the SteamVR system's working area was calibrated, which determined the location of its coordinate system. The controller from the HTC VIVE set was used for the calibration. The origin of the $Y_v$ axis, denoting height, was determined at floor level by measuring the height of the controller placed on it. The $X_v$ and $Z_v$ axes were set parallel to the boundaries of the rectangular working area. This area was

determined by measuring the position of the controller at the four corner points so that the origin of the SteamVR coordinate system ($O_v$) coincides with the centre of the working area ($O_r$). In turn, the $X_v$ and $Z_v$ axes should be parallel to their limits.

During preliminary measurements, however, to appeared that the SteamVR coordinate system did not coincide with the assumed coordinate system. The differences concerned both the shift between the origins ($X_c$, $Y_c$, $Z_c$) as well as the rotations ($A_c$, $B_c$, $\Gamma_c$); as a result, the directions of the individual axes of both systems did not coincide (Figure 4). Such a result was expected, as similar problems were noted by other authors of the SteamVR system research [26,27].

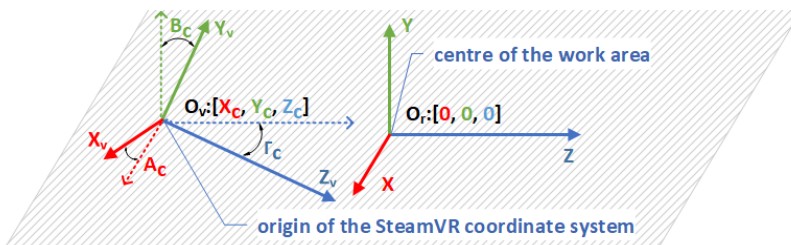

**Figure 4.** Translation and rotation between the real coordinate system and the SteamVR system. The hatched area represents the floor.

The values of translation and rotation shown in the above diagram, once the calibration was corrected, did not exceed a few centimetres and a few degrees for each respective axis. The translation and rotation values for the individual axes changed when the SteamVR was restarted, or when synchronisation between the base stations was lost. The resulting errors could be compensated by additional calibration before using the SteamVR system. This issue is discussed in greater detail throughout the paper.

Data acquisition consisted of manually setting the launcher position on a tripod in accordance with the guidelines calculated by the program. After setting, the user started the acquisition of results containing information about the position and orientation of the tracked objects, such as the tested tracker, the HMD goggles, and the base stations.

The measurement method described above is an extension of the method used in [25], where the tripod was also set at subsequent points in the room. In the method described here, however, a tripod head rotation element has been added to assess potential problems with tracker visibility more accurately. In work on motion tracking systems, measurements with an industrial robot arm are often used [21,28], but this method makes it difficult to perform tests in a larger working area.

## 3. Measurement Results

For each position and orientation of the launcher, 50,000 data samples were recorded for all the SteamVR devices (tracker, HMD goggles, and base stations). However, the research concerned the custom tracker, so the presented results are limited to this device only. Nevertheless, the measurement results for the remaining components of the system turned out to be useful in interpreting the obtained results. The two main parameters of the tracker have been defined, which are the trueness and precision of the measurement. The results are shown below for position and orientation, separately.

### 3.1. Position

The position of the tracker changed when the tripod was moved to the next measuring point, as well as with the change of the launcher orientation, as the tracker's centre was not in the tripod head's axis of rotation. The attachment point of the launcher was selected to correspond to the place where the launcher rests against the user's arm in the actual application. The position of the tracker at each of the nine points where the tripod was placed should have values belonging to five circles, the radius of which will depend on the

angle of the tripod inclination $\theta$. The average values of the tracker positions are shown in the XZ plane in Figure 5. The chart also shows circles whose parameters were determined using the least-squares method.

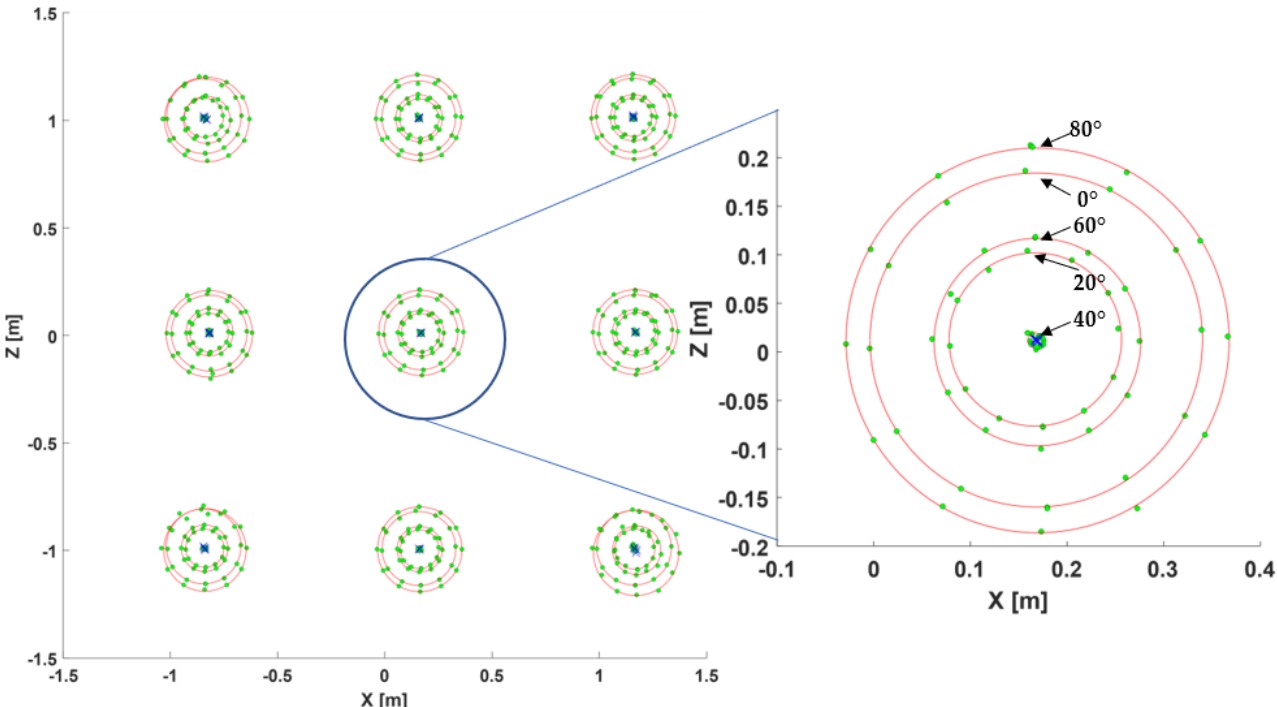

**Figure 5.** Tracker positions during measurements (top view). The dots indicate the tracker's position, the red lines represent the circles with a radius determined from the results, and the crosses indicate their centres.

The chart shows four circles for each tripod position. The fifth circle for the elevation angle of $40°$ is almost invisible, as the average radius of the circles is about 9 mm. The maximum differences between the radii of the circles for the same elevation angle are about 5 mm for a $0$–$40°$ elevation, and increase to almost 10 mm for an $80°$ elevation. In part, these values may be due to an inaccurate setting of angle $\theta$ on the tripod head.

The circles projected onto the XZ plane should be concentric. However, especially at the corner points of the working space, the centres of the circles are shifted. The greatest shift values were observed near the base stations, where the centres of the circles are 27 mm apart near the BS1 stations, and 22 mm near the BS2 stations. The circles in the centre of the working space are closest to one another, with the distance between their centres not exceeding 4 mm.

The results show a systematic error of approximately 160 mm for the X coordinate and 12 mm for the Z coordinate. This error results from the shift of the SteamVR coordinate system relative to the centre of the working space. It can be eliminated at the VR application stage. Often, however, there is no need for this, as the same shift is observed for the positions of the HMD goggles which are positioned in the same coordinate system. As a result, when the user sees the virtual world through the HMD goggles, such a shift is difficult to notice, and has no impact on the working environment.

In addition to the parameters of circles discussed above, there is also the Y coordinate of the centre of the circle, and a vector normal to the plane of the circle, which describes its orientation in space. A graphical representation of the above parameters is presented in Figure 6, showing the data projected onto the XY plane (side view).

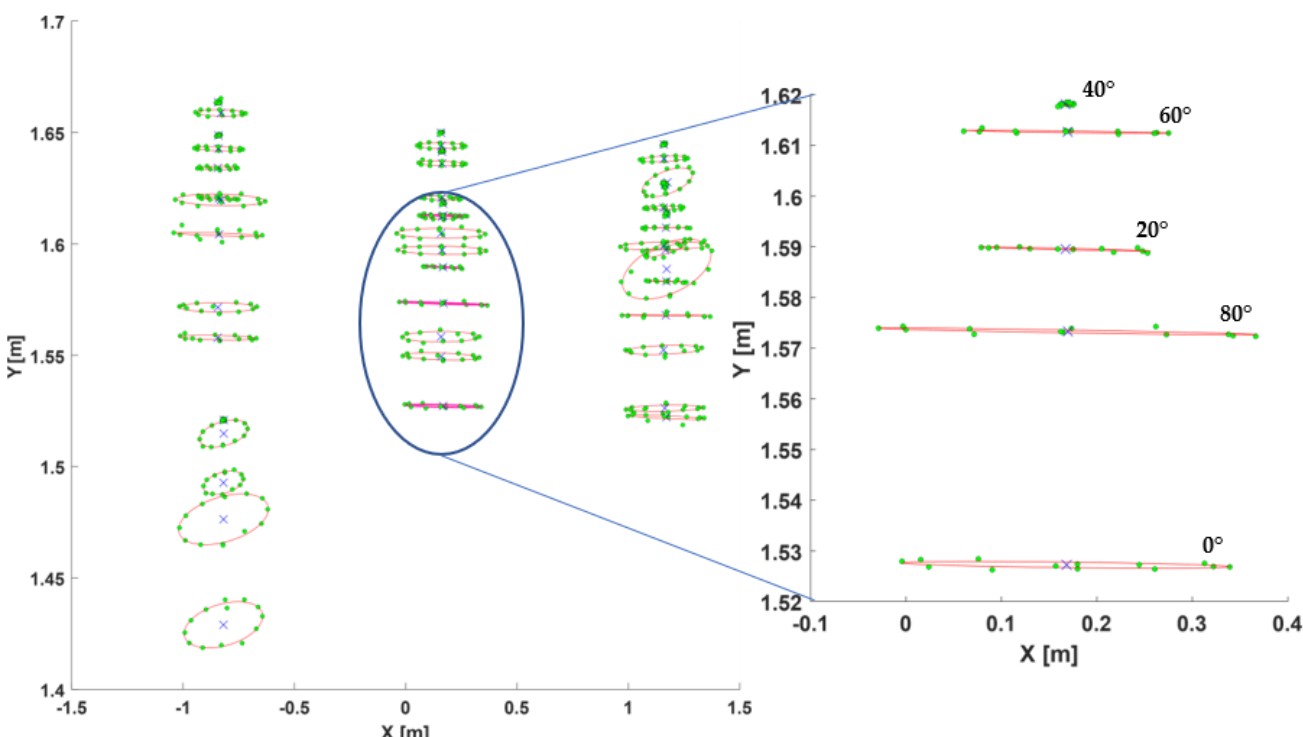

**Figure 6.** Tracker positions during measurements (side view). The dots indicate the tracker's position, the red lines represent the circles with a radius determined from the results, and the crosses indicate their centres.

For points with a greater value of X, the readout for the tracker's height decreases, which indicates the rotation of the SteamVR coordinate system with respect to the room. As a result, the "virtual" floor plane in the SteamVR system is tilted relative to the actual floor. The tilt angle is 0.5°, which, for a 2 by 2 m working space, translates into a height change of less than 20 mm between the extreme points. These values are small, and can be compensated for in a virtual environment. There were no significant differences between the deviation of the height of the circles at different elevation angles.

The rotation of the planes to which the individual circles belong was also observed. The mean difference between the reference plane normal (0, 1, 0) and the determined circle plane norm is 1° with a standard deviation of 1°. The worst results of deviation from the horizontal direction (3.5°) were recorded for the measurements at the point where x = −1, z = 0. At the same time, the standard deviation at this point remains small, which confirms the rotation of the SteamVR reference frame. The best results were achieved in the centre of the room, where the mean inclination between the plane of the circles and the plane of the floor was 0.21° with a standard deviation of 0.05°.

### 3.1.1. Position Precision

For each position of the launcher, the standard deviation (*SD*) was determined, which represents the precision of the measurement. The summary of the results with their histograms is shown in Figure 7, which shows the fragments where the position of the tripod was constant, and the launcher orientation was changed.

This section may be divided by subheadings. It should provide a concise and precise description of the experimental results and their interpretation, as well as the experimental conclusions that can be drawn.

Among all 585 analysed results, only three tracker positions were characterised by a standard deviation slightly greater than the assumed 1 mm. The vast majority of the results were characterised by a standard deviation of less than 0.6 mm for the X and Z axes, and less than 0.1 mm for the Y axis. Thus, it can be concluded that, in terms of measurement

precision, the tracker works properly. Thanks to this, in the VR environment, the user should not observe the instability of related virtual objects.

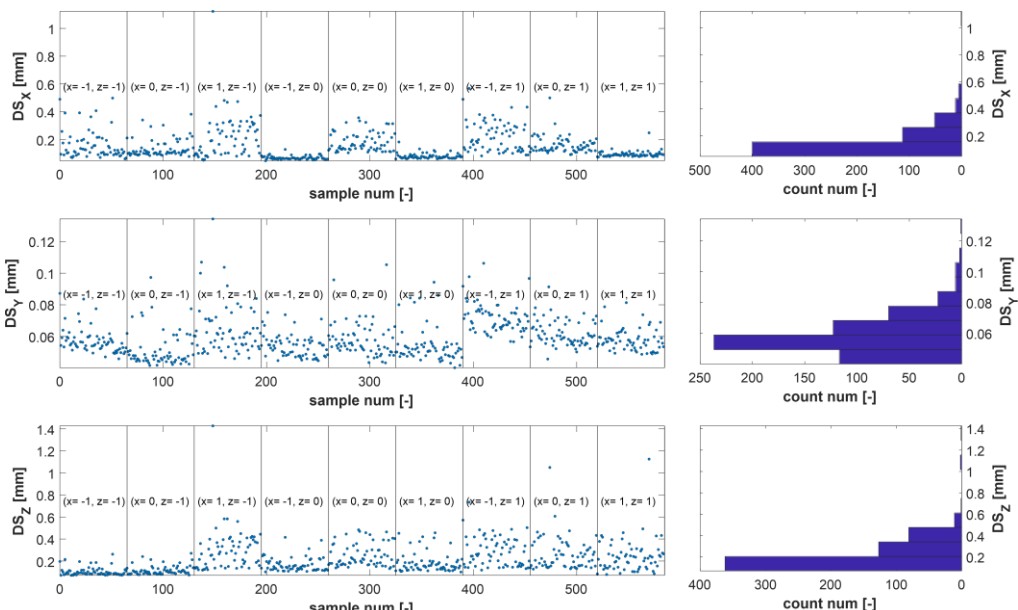

**Figure 7.** The precision of the tracker position measurements for the individual components (**left**), and their histograms (**right**).

### 3.1.2. Position Trueness

The measurement trueness, quantified by the measurement deviation ($\Delta$), was calculated as the difference between the mean measured value and the true value. The latter was calculated based on the tripod orientation and position data, as well as the parameters of the tripod head. Constant correction factors were included in the calculations to compensate for shifts in the SteamVR coordinate system. The results are shown in Figure 8.

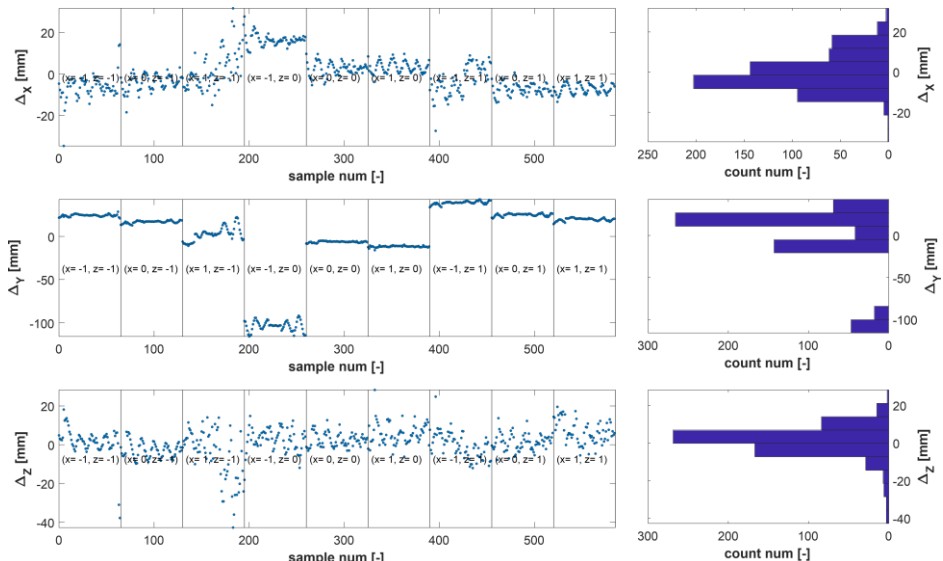

**Figure 8.** The trueness of the tracker position measurements for the individual components (**left**) and their histograms (**right**).

In most tests, the measurement deviation of the position components does not exceed $\pm20$ mm. The exception is the fourth group of samples corresponding to the point x = −1,

z = 0. The existence of such a large deviation results from the shift of the SteamVR coordinate system, and does not preclude the application possibilities of the developed device. If only this fragment of data was analysed, after calculating the adequate values of the shifts between the coordinate systems, the systematic measurement error would not exceed 25 mm at point x = −1, z = 0. A similar analysis could be extended to other parts of the chart above. For example, for a tripod in the centre of the working area, the measurement deviation would be less than 10 mm after offset correction.

### 3.2. Orientation

For a clearer presentation of the orientation results, they were plotted as a set of fixed-length vectors representing the direction in which the launcher was pointing. Each of the vectors was parallel to the axis of the launcher, and shifted downwards by an angle of 10°, which results from the mounting angle of the tracker (Figure 9). The desired value of the tripod position in the real coordinate system was taken as the starting point of the vectors.

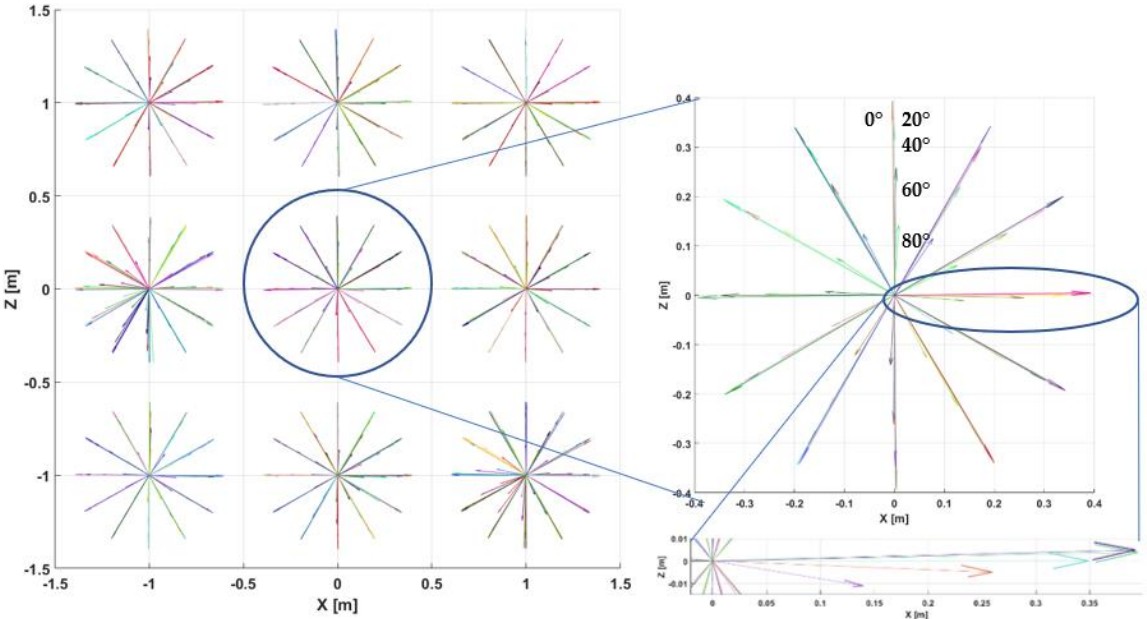

**Figure 9.** Vectors indicating the direction in which the launcher was facing during the measurements (top view).

The data presented in the figure above show that the tracker orientation was determined correctly. The vectors at each point were evenly distributed around the circumference of the circle; however, as in the case of the position, some shifts were observed. All vectors were rotated anticlockwise about the Y axis by approximately 1°. Moreover, along with the increase in the value of the elevation angle (shorter projection of the vector onto the XZ plane), the shift angle changed anticlockwise, increasing the measurement's systematic error. The reason is probably the rotation between the real coordinate system and the SteamVR system. The side view of the vectors shown in Figure 10 is slightly less readable.

As in the case of the analogous position analysis, here the results from several tripod positions also overlapped. However, it can be seen that the vectors were sorted into groups corresponding to the successive values of the elevation angle. The angle between the vectors for successive values of $\varphi$ at the equal value of the angle $\theta$ is approximately 20°, which corresponds to the actual changes. All vectors were rotated downwards by 10°, which coincided with the mounting angle of the tracker.

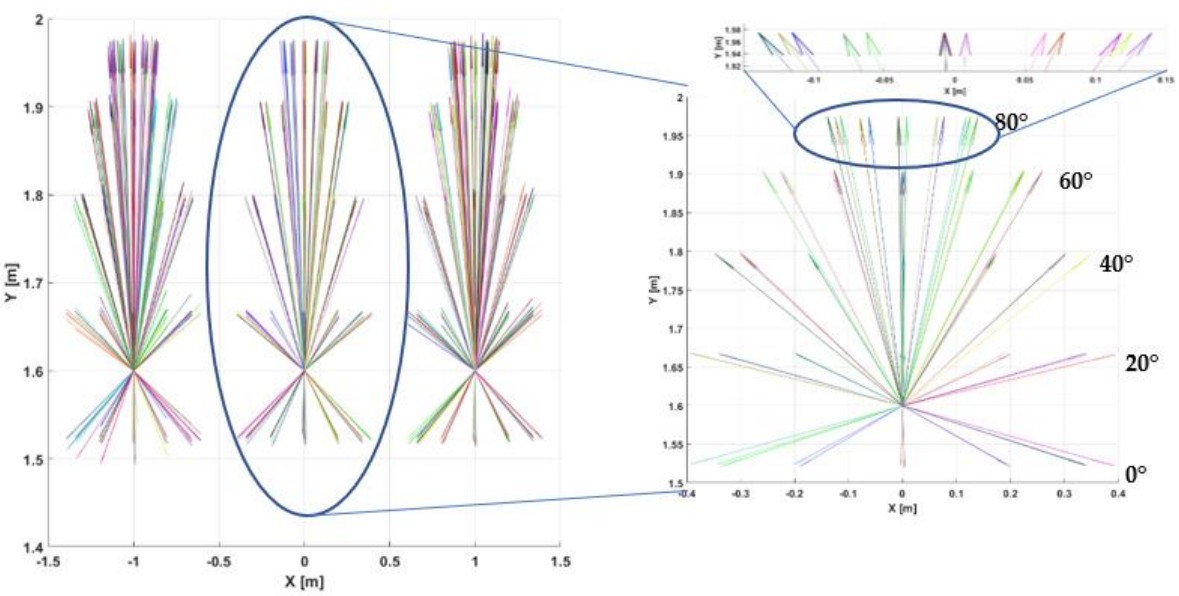

**Figure 10.** Vectors indicating the direction in which the launcher was facing during the measurements (side view).

### 3.2.1. Orientation Precision

The precision of the orientation measurement, as in the case of the position, was defined as the standard deviation of the results determined for each of the components of the orientation vector. The results are shown in Figure 11.

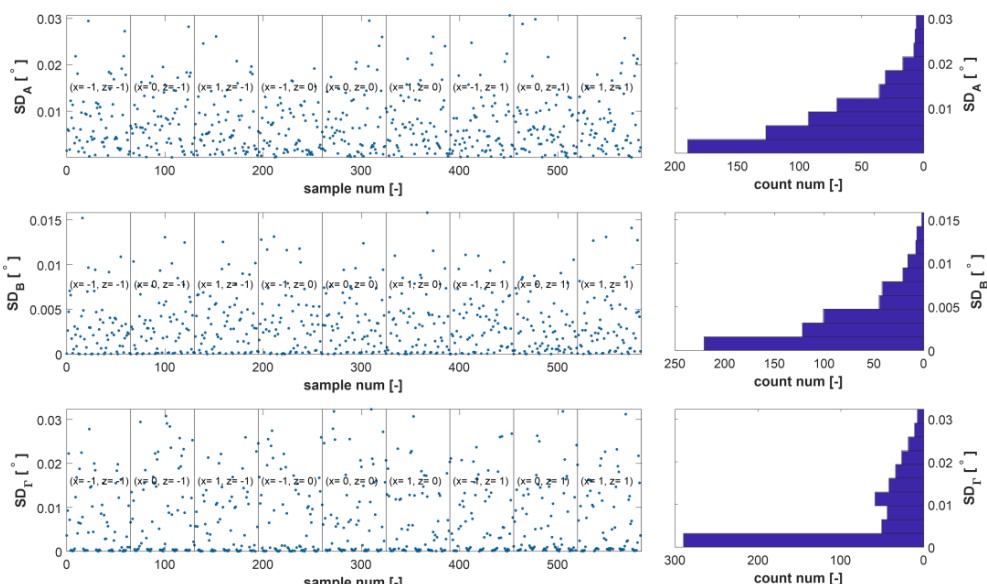

**Figure 11.** The precision of the tracker orientation measurements for the individual components (**left**) and their histograms (**right**).

The values of the standard deviation determined for the A and Γ axes are below $0.03°$, and for the B axis below $0.015°$. Thus, the precision of the orientation measurements for the tested tracker meets the requirements, as the random error does not exceed $0.1°$. The graph above shows no correlation between the precision of the orientation measurements and the tripod's position or orientation.

### 3.2.2. Orientation Trueness

The measurement trueness was analysed differently than in the case of the previous parameters, as the description of the orientation in the form of Euler angles is ambiguous. Very similar launcher orientations can be described by different sets of angles ($\alpha$, $\beta$, $\gamma$). Thus, the calculated measurement deviation of the individual components could suggest significant discrepancies, while different datasets describe a similar orientation. In the SteamVR system, the orientation is recorded in the form of quaternions, which accurately describe the differences between the true and measured orientation (the systematic error of the orientation measurement). However, this method is difficult to interpret, hence we converted the quaternions to Euler angles. During the measurements, both forms of the orientation description were noted, so quaternions were used to determine the total systematic error of the orientation measurement.

Based on the data on the orientation of the tripod head (angles $\varphi$ and $\theta$), a matrix of quaternions describing the true orientation was determined. Then, the angular difference between the mean value of the measured and true orientation was determined. The functions quaternion ( ) and dist ( ) of the Matlab 2019b environment were used to calculate the angular difference and convert the Euler angles to quaternions. The calculation results are shown in Figure 12.

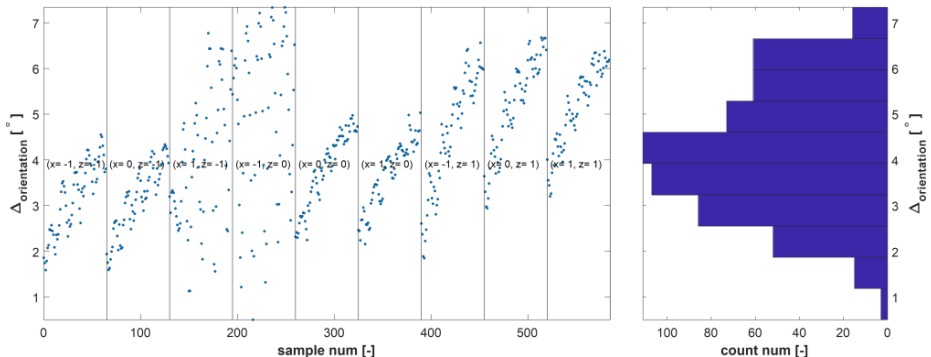

**Figure 12.** The trueness of the tracker orientation measurements (**left**) and their histograms (**right**).

The average measurement deviation of the tested tracker orientation was about $4°$. This value was greater than assumed; however, the repeated correlation between the trueness and the orientation of the tripod head, which repeats for each tripod position, proved the rotation of the SteamVR coordinate system in relation to the real coordinate system. The rotations were confirmed by similar kinds of changes in the measurement deviation for the HMD goggles placed on the same launcher. Thus, it was possible to reduce the systematic error of the orientation measurement by introducing a correction in the virtual environment.

## 4. Discussion

The conducted measurements assumed the analysis of 585 different tracker positions in the working space. The tested tracker was continuously tracked by the SteamVR system while recording data from all measurement points. Taking into account that the measurement setup realistically reflects the conditions in which the tested device is to be used, it can be concluded that the developed tracker is free from the disadvantages of commercial devices, which in similar conditions had problems with tracking due to being covered by the launcher or the user's head.

Both the position and orientation of the tested tracker were measured with a precision that met the requirements for a motion tracking system for a virtual shooting simulator application. These parameters are extremely important, as they determine the correct course of the basic task for the user, which is to keep the object in the launcher's sight. This task would be impossible if the tracker and associated launcher were not positioned

precisely. Especially important is the tracker orientation, for which the average value of the standard deviation is more than two times lower than the assumed limit value. There was a shift and rotation between the SteamVR coordinate system where the tracked objects were positioned and the actual system in the room. This meant that both the position and orientation of the tracker were measured with insufficient correctness. The resulting measurement error was, however, a systematic error and, as such, it can be significantly reduced by introducing appropriate corrections. These, in turn, can be introduced in the virtual environment itself, taking into account the fact that similar errors concern the HMD goggles through which the user observes the virtual world. Therefore, the shifts and rotations observed by them should not have a large impact on the course of the simulation.

Total values of systematic and random errors of measurement were determined for position and orientation. Total values of random errors of position and orientation measurement were determined as the RMS value of the individual components. The total systematic error of position measurement was determined as the Euclidean distance calculated from the individual components. For orientation, as given in the previous chapter, the values of the components of systematic error were not determined, but only the total error. The summary of the mean values of all the tested parameters, together with the standard deviations, is presented in Table 1.

**Table 1.** Summary of the tracker measurement results.

| | Position | | | | | Orientation | | | |
| --- | --- | --- | --- | --- | --- | --- | --- | --- | --- |
| | Systematic Error [mm] | | Random Error [mm] | | | Systematic Error [°] | | Random Error [°] | |
| | Mean | Standard Deviation | Mean | Standard Deviation | | Mean | Standard Deviation | Mean | Standard Deviation |
| **X** | 7.0 | 5.4 | 0.14 | 0.10 | **A** | - | - | 0.007 | 0.006 |
| **Y** | 28.1 | 28.6 | 0.06 | 0.01 | **B** | - | - | 0.003 | 0.003 |
| **Z** | 5.6 | 5.5 | 0.21 | 0.13 | **Γ** | - | - | 0.007 | 0.008 |
| **Total** | 31.0 | 28.2 | 0.15 | 0.09 | **Total** | 4.2 | 1.4 | 0.008 | 0.004 |

Based on the obtained results, the centre point of the working area was selected as the optimal place to set up a person training with a launcher. The next stage of the work will be tests of the system with a user, which will verify the possibility of introducing corrections improving the accuracy of the measurements, and thus will give the final answer as to the application possibilities of the developed tracker.

The obtained results concerning the correctness and precision of the measurement for the tested custom-made device are comparable with the results obtained for commercial devices [21,26,27]. Moreover, the tested tracker, in comparison with the commercial device, showed uninterrupted tracking of the launcher [24]. This confirms that it is possible and advisable to design and construct SteamVR devices with the shape and arrangement of sensors optimized to the tracking of a given object.

## 5. Conclusions

Custom-made SteamVR Trackers are an interesting addition to the SteamVR system. In many applications, commercially available trackers are sufficient, however, when these devices do not fulfill their tasks, for example due to the large size of the tracked object, a custom-made tracker can be used. The authors of the previously cited works mention the use of several trackers to track one object as one of the possibilities of improving the system parameters. Thanks to the custom-made approach, the entire object could be covered with sensors, thus acting as a tracker, which, due to its dimensions, would be well tracked. This can be an important issue, as the number of SteamVR devices operating simultaneously is limited. The results described in this paper confirm that, in terms of measurement precision

and correctness, custom made trackers work similarly to commercial devices and, thanks to optimization, they can track motion when commercial devices would be obscured.

The next planned stage in the development of the device described in the article will be to test it together with a reference commercial device in the target shooting simulator application. Based on the results obtained and the assessment by users performing typical training tasks, the quality of this device will be verified in comparison with a commercial device.

**Funding:** This research was funded by Military University of Technology; grant number UGB/22-831/2021/WAT.

**Informed Consent Statement:** Not applicable; did not involve humans.

**Acknowledgments:** I would like to thank Norbert Pałka for his help in technical editing, language editing, and proofreading.

**Conflicts of Interest:** The author declares no conflict of interest.

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
