# Peer review of "Evaluation of the SteamVR Motion Tracking System with a Custom-Made Tracker"

_applsci, doi:10.3390/app11146390_

Round 1

Reviewer 1 Report

The evaluation of a motion tracker for an air-defence portable VR training system is an important and timely topic. Overall, it is an interesting manuscript however it presents some weaknesses that need to be addressed prior to publication consideration. Following, concrete comments for required changes/corrections and recommended additions are provided:

The main weakness of the article is the briefness of the Introduction and the weak literature review. As a result the article cites a striking low number of references.

Statements such as “Virtual reality is used, among others, in medicine, sport, and in military applications, where the typical use is for shooting simulators” (l26-7) should be accompanied and supported by relevant literature sources.

The same applies to the sentence on l35-7 “The transfer value of virtual training simulators is influenced by several factors, such as immersion, i.e., the impression of being present in the virtual world, fidelity of the real-world representation, and the level of user acceptance of a given simulator.”: this claim needs to be confirmed by bibliography.

On l46-47, the author describes the available motion tracking systems in the market, a vital information. However, the only reference provided is from 12 years ago. Obviously more current data is paramount for the accuracy and validity of information and following arguments.

L53-4: Please add reference(s) on the lighthouse solution.

L54-6: Please add more information and reference(s) on the used devices. To most readers, Steam VR is an online software platform, not a hardware component.

L57-8: Please add reference(s) on PIORUN.

L87: Were there other similar attempts or efforts in this sector linking VR with motion trackers or other sensors for simulation fidelity? A brief literature review section prior to the setup would be very useful.

L162-3 “It turned out, however, that the SteamVR coordinate system did not coincide with the assumed coordinate system”: Please provide more information how this conclusion was reached and how the differences were determined. If you are referring to information that follows, please state this explicitly so that readers can follow your line of thought and actions.

The author needs to explain and justify the selection of the specific system evaluation methods. Did he follow a specific paradigm based on previous works or was it a newly developed routine?

L340: The discussion section is also very short and the article ends abruptly. Findings should be compared with other similar efforts / projects / research studies.

Furthermore, I suggest adding a Conclusion section where the “big picture” of this work in the larger context is explained, the next steps will be defined and the contribution of the manuscript to the field is highlighted.

L378 - References: All references should be checked thoroughly as several of them such as [4] (url missing) and [6] (journal & doi missing) need corrections to adhere to the journal’s standards.

Reviewer 2 Report

The authors claim to evaluate a custom SteamVR tracker for man-portable air-defence training systems. The precision and orientation results are adequate for a motion tracking system for a virtual shooting simulator application. The paper is interesting and the research method is good, however there are some issues with the format and the lack of sufficient background information.

Please clearly define how you are evaluating the precision of your system. Did you compare the data with another SteamVR tracker or with a certified optoelectronic solution? How accurate can you manually set the angle of a tripod head? Did you use a digital measurement system?

Regarding formatting:

  • check figure alignment and position
  • figures are overlapping with the caption
  • figures with two-three images should be separated and highlighted in the caption the difference between them

Round 2

Reviewer 1 Report

The author has addressed satisfactorily all previously mentioned weaknesses hereby improving significantly the quality and readability of the manuscript. 

Congratulations!

Author Response

Thank you for your review.

Reviewer 2 Report

Dear authors,

My comment regarding Point 6 was that you left that section from the template. However, the changes you've made are useful as the section is easier to follow.

The format for the table is still wrong (or so it appears to me). Please check https://www.mdpi.com/journal/applsci/instructions

Lastly, the sentences from 426-429 should have the same font as the body, not as the Funding.

Author Response

Thank you for your review. I am sending the replies to your comments in the attachment
